# Poly(ADP-Ribose) Polymerase Inhibition as a Promising Approach for Hepatocellular Carcinoma Therapy

**DOI:** 10.3390/cancers14153806

**Published:** 2022-08-05

**Authors:** Alexia Paturel, Janet Hall, Isabelle Chemin

**Affiliations:** Université de Lyon, Université Claude Bernard Lyon 1, INSERM, CNRS, Centre Léon Bérard, Centre De Recherche En Cancérologie De Lyon, 69008 Lyon, France

**Keywords:** liver cancer, hepatocellular carcinoma (HCC), hepatitis B virus (HBV) X protein, poly(ADP)ribose polymerases (PARP)

## Abstract

**Simple Summary:**

Liver cancer has very high incidence and mortality rates, making it a major public health problem. Indeed, the available treatments are not numerous, and few strategies exist for patients with advanced cancers. Thus, there is an urgent need to search for new treatment targets. This review provides an overview of one potential target, poly(ADP)ribose polymerase 1, a protein involved in DNA repair pathways and its expression profiles in liver cancer. Inhibition of this protein could potentiate the effects of current treatments and improve therapeutic outcomes for patients.

**Abstract:**

Primary liver cancer is the sixth most common cancer in men and seventh in women, with hepatocellular carcinoma (HCC) being the most common form (75–85% of primary liver cancer cases) and the most frequent etiology being viral infections (HBV and HCV). In 2020, mortality represented 92% of the incidence—830,180 deaths for 905,677 new cases. Few treatment options exist for advanced or terminal-stage HCC, which will receive systemic therapy or palliative care. Although radiotherapy is used in the treatment of many cancers, it is currently not the treatment of choice for HCC, except in the palliative setting. However, as radiosensitizing drugs, such as inhibitors of DNA repair enzymes, could potentiate the effects of RT in HCC by exploiting the modulation of DNA repair processes found in this tumour type, RT and such drugs could provide a treatment option for HCC. In this review, we provide an overview of PARP1 involvement in DNA damage repair pathway and discuss its potential implication in HCC. In addition, the use of PARP inhibitors and PARP decoys is described for the treatment of HCC and, in particular, in HBV-related HCC.

## 1. Introduction

Primary liver cancer is the sixth most frequently found cancer in men and seventh in women, with hepatocellular carcinoma (HCC) being the most common form (75–85% of primary liver cancer cases). In 2020, the ratio of mortality to incidence was 830,180 deaths for 905,677 new cases [1]. The most frequent etiologies are infection with hepatitis B (HBV) and C (HCV) viruses, alcohol intake and the ingestion of the fungal metabolite aflatoxin B1. In addition, the cooperation of risk factors is well described in the pathogenesis of HCC. In fact, exposure to aflatoxin B1 acts synergistically with HBV and increases the relative risk of developing HCC from 11.6% with HBV infection alone to 64% for HBV infection combined with aflatoxin exposure [2]. The contribution of different etiologies varies in different regions of the world, but it is estimated that more than 80% of HCC cases are attributed to viral infections. Treatments for early stage HCC include ablation, resection and transplantation, while intermediate-stage HCC requires transarterial chemoembolization (TACE). Few options exist for advanced or terminal-stage HCC, which will receive systemic therapy or palliative care [3]. Despite these different therapies, the 5-year survival rate was 24.3% for patients diagnosed with HCC between 2004 and 2006 [4]. One of the contributing factors to this poor survival is that HCC is recognized as one of the most chemo-resistant tumor types, and, as a consequence, few treatment options exist. For instance, it has taken almost 30 years for the approval of Sorafenib, an oral multi-TK that is the current standard of care for patients with advanced HCC. Thus, there is an interest in identifying drugs to improve patient prognosis.

To date, radiation therapy is not used in the treatment of HCC, except in rare exceptions, such as the palliative setting. Many advances in this field, such as improved delivery, suggest that RT should be re-evaluated as a potential therapy for HCC. In particular, hypofractionated sterotactic body RT (SBRT) (reviewed in Refs [5,6]) as monotherapy or combined with either liver-directed therapies or radiosensitizing drugs, such as inhibitors of DNA repair enzymes, should be investigated. Indeed, there is considerable interest in exploiting tumor-specific imbalances and deficiencies in DNA damage repair (DDR) caused, for instance, by mutations in DNA repair genes and the subsequent modulation of DDR capacity by the use of small molecule inhibitors of DNA repair enzymes for therapeutic benefit.

Inhibitors of poly(ADP)ribose polymerases (PARP) (PARPi) that are involved in the regulation of many cellular processes are of particular interest. Poly(ADP-ribosyl)ation (PARylation) is a post-translational modification that modulates many cellular processes, including transcription and chromatin dynamics, in addition to playing a key role in DDR. Among the 17 members of the PARP family, PARP1 is the most abundant family member and is estimated to be responsible for ~85% of the total cellular poly(ADP-ribose) (PAR) synthesis [7]. PARP1 plays a role in several DDR pathways, including base excision repair (BER), homologous recombination (HR), classical non-homologous end joining (NHEJ) and alternative NHEJ (Alt-NHEJ), also known as microhomology-mediated end joining (MMEJ) [8]. In addition, there is evidence that PARP2 and PARP3 also play roles in DDR [8]. Thus, the inhibition of PARP activity through the use of inhibitors that target the structurally similar catalytic domain of these proteins would be expected to have an impact on how cells respond to DNA damage generated both intrinsically, through, for instance, oxidative stress, or extrinsically, from chemotherapy or radiotherapy.

This impact can be attenuated by the mechanism of synthetic lethality that corresponds to cell death obtained by the synergistic action of two non-lethal events, resulting from, for example, the accumulation of unrepaired DNA strand breaks because of the loss of multiple DDR pathways, leading to cell death. One example of synthetic lethality that was preclinically demonstrated in 2005 and is now clinically exploited is the use of PARP inhibitors in breast and ovarian cancers that have developed in BRCA 1 (BReast Cancer 1) or BRCA 2 (BReast Cancer 2) genetic backgrounds. Such tumors have “lost” the second BRCA allele and are deficient in the BRCA encoded enzymes involved in one of the major DNA pathways for DNA double strand breaks (DSB) in replicating tumor cells. The action of PARP inhibitors will thus lead to an accumulation of DNA DSB and tumor cell death. Four small molecule catalytic PARPi have now been approved by the U.S. Food and Drug Administration (FDA) and the European Medicines Agency (EMA) for various cancers, including ovarian, breast, fallopian tube or primary peritoneal cancers [9], with several others in different stages of development and clinical assessment. In HCC, several clinical studies have been conducted with PARPi, including one that included a combination with the anti-cancer agent Temozolomide [10]. However, to date, no clinical studies have assessed their impact in combination with RT in HCC.

This article will briefly review the involvement of PARP in DNA repair pathways and discuss the mechanism of action of PARPi. PARP is an Achilles heel of HCC. The preclinical results of PARPi in combination with radiotherapy for the treatment of HCC will be reviewed, as well as other possibilities for targeting this key protein.

## 2. PARP1 as a Key Protein in DNA Damaged Repair

Many different types of DNA damage are induced by cancer treatments. For example, it is estimated that irradiation of a cell with a dose of 1 Gy leads to the generation of 10,000 different base alterations (loss, oxidation, etc.), 1000 single strand breaks (SSBs) and 40 double strand breaks (DSBs) [11]. While DNA DSBs are less numerous, they are more toxic than SSBs [12]. In the presence of a DSB, several cellular outcomes are possible: (1) if the DSB is repaired quickly, the cell continues its cycle and divides; (2) if the break is not repaired immediately, the cell will stop the division cycle until the repair is completed; and (3) if the DSB is not repairable or is misrepaired, it will lead to cell death or mutagenesis.

DSBs are the most difficult DNA lesions to repair. Indeed, mammalian cells have evolved several repair mechanisms for their repair: HR, NHEJ or MMEJ. However, DNA DSBs are mainly repaired through two pathways: HR or NHEJ. These processes can be accurate and allow cell survival without genetic consequences or be prone to errors, leading to the generation of mutations or chromosomal aberrations. In fact, unlike NHEJ that joins ends together without a template, HR is an error-free pathway that utilizes the sister chromatid as template. Whereas the NHEJ repair pathway works particularly during G1-G0 and may also function during the S phase of the cell cycle, the HR pathway is used in G2-S phases [13].

The repair of SSBs occurs via the later stages of the BER pathway and involves the recognition of the DNA break by PARP. Binding of PARP1 to the SSB leads to its activation and to the PARylation of various nuclear proteins, including PARP itself [14].

The rapid binding of PARP1 to DNA strand breaks through its N-terminal Zinc finger motifs is critical not only for the resealing of DNA SSBs during BER but also for the repair of topoisomerase I cleavage complexes and for DSB repair through the modulation of the activity of proteins involved in NHEJ. PARP1 is also involved in the MMEJ DSB repair pathway. The resulting conformational change on binding the damaged DNA activates PARP1’s C-terminal catalytic domain to hydrolyze NAD+ and produces linear and branched PAR chains that can result in PARP1 auto-ribosylation and the trans-ribosylation of other target proteins. The polymers form a molecular scaffold that can lead to protein redistribution either away from the DNA because of charge repulsion between the negatively charged polymers or into multi-protein repair complexes via the recruitment of PAR-binding motif-containing proteins to sites near or at DNA damage. For instance, the autoPARylation of PARP1 promotes the recruitment of the DNA repair factor XRCC1 that is necessary for the resealing of DNA SSBs during BER. It is also essential for PARP1’s dissociation from DNA, which is required for the completion of DNA repair.

PARPi exert cytotoxic effects on cancer cells through two main mechanisms: inhibition of PARP catalytic activity and PARP scavenging, whereby the PARP protein inactivated by the PARPi does not easily dissociate from DNA damage, preventing DNA repair, replication and transcription and finally leading to apoptosis and/or cell death [15]. Indeed, it has been shown that these PARP complexes are more cytotoxic to cells than unrepaired SSBs caused by the absence of PARP proteins, and as such, PARPi have been proposed to act as PARP poisons [15]. 

## 3. PARP in HCC

A prerequisite for the clinical use of PARPi in different tumor types is PARP expression. Many studies evaluating PARP transcript and protein expression profiles have focused on PARP1, which, as discussed above, is responsible for about 85% of cellular PARylation activity and for which good probes and antibodies are available. PARP1 mRNA levels are upregulated in several cancer types, including HCC [16,17]; however, the expression of PARP1 protein did not parallel these changes in all the tissue panels examined [17,18,19,20]. PARP1 and PARP2 mRNA levels were also found to be upregulated in liver cancer cell lines compared to primary human hepatocytes (PHH). In addition, a correlation was found between the PARP1 mRNA and protein levels. However, no correlation was found between the protein level of PARP1 and its enzymatic activity in cell lines, suggesting that many factors may influence PARP expression and activity [21]. 

This raises the question of the involvement of PARP1 in HCC tumorigenesis but also in resistance and sensitivity to current therapies. Firstly, a significant correlation was found between cells positive for the proliferation marker Ki-67 and the relative activity of PARP in HCC patients [19]. Indeed, several studies demonstrate a correlation between PARP1 activity and hepatocyte proliferation in vivo and in vitro. For instance, Ju et al. show that in PARP1 knockout mice, hepatic regeneration after partial hepatectomy is impaired. In addition, their work showed that PARP1 knockout inhibits cell-cycle regulatory proteins, such as cyclin B1 and D1. This inhibition could be achieved by repressing the activity of the YAP [22].

Secondly, a binding site of PARP1 has been identified in a wide range of proteins, including nuclear factor-kappa B (NF-κB) [23]. Subsequently, it was found that inhibition of PARP1 led to an improvement in inflammatory disorders via suppression of NF-κB [24] and that PARPi were a means of targeting cancer cell apoptosis through their effect on the NF-κB signaling pathway in HCC [25]. In addition, Hassa and colleagues showed that PARP-1 acts synergistically with p300 and plays an essential regulatory role in NF-κB-dependent gene expression [26]. Therefore, it is tempting to speculate that PARP overexpression in HCC is a carcinogenic factor due to its anti-apoptotic effect through the NF-κB signaling pathway.

In some types of cancer models, it has been reported that PARP1 deletion contributes to a defective activation of transcription factors that play a key role in tumor development, such as NF-kB or hypoxia inducible factor (HIF) [27]. Other studies have revealed that the absence of PARP1 modulates HIF1 accumulation by reducing both nitric oxide and oxidative stress. These results suggest that PARP1 is involved in the fine tuning of the HIF-mediated hypoxic response in vivo [28]. However, to date, no study has confirmed these effects between HIF and PARP1 in the liver.

Finally, PARP1 expression is closely associated with β-catenin accumulation and promotes the transcription of numerous oncogenes, such as c-Myc and Cyclin D1 [29]. Similarly, PARP1 inhibition has been shown to inhibit β-catenin signaling and its downstream components, such as c-Myc, cyclin D1 and MMPs [30]. Altogether, this evidence suggests a potential role of PARP1 in HCC carcinogenesis. Its overexpression may not only confer a survival advantage to cancer cells but may also underlie cancer initiation through its effect on signaling pathways, such as HIF, NF-κB or β-catenin. However, it is very important to underline that there are too few studies to understand the mechanisms underlying PARP as a potential driver of HCC.

## 4. PARP Inhibitors or PARP Baits for HCC Therapy

In a previous study, we showed in tumor tissues from HBV, HCV or alcohol-associated etiologies reduced levels of the DNA damage signaling molecule H2AX compared to peritumoral and control tissues, a profile also reported during the progression of control to tumor tissue in the breast [17]. However, in the same panel of liver tissues, the reverse pattern was found for the surrogate DNA damage marker gammaH2AX with higher levels in the tumor tissues compared to the peritumoral and control tissues [17]. Evert et al. [31] also reported an elevated level of gammaH2AX in HCC tumor tissue. Taken together, these two studies are indicative of an imbalance of DSB repair in HCC that could be further exploited by the use of PARP inhibitors. PARPi has a potential for use in HCC treatment through at least two mechanisms: first, by the catalytic inhibition of PARP activity after its binding to DNA damage and second, as a consequence of this inactivation of PARP activity, by the trapping of the PARP protein on DNA damage that would block replication. As discussed above, PARP1 is the most abundant cellular PARP protein, and the “trapped” PARP1 would need to be removed by other cellular repair processes to avoid the generation of replicative stress. The first results in the literature using liver cancer cell lines as a model system showed that out of seven cell lines, four were sensitive to the clonogenic cell killing effects of Veliparib (ABT-888), given as a single-dose exposure [21]. It was later recognized that the PARP inhibitors developed have different PARP1 trapping efficiencies, with Talazoparib having the highest trapping efficiency (talazoparib >> niraparib > olaparib = rucaparib >> veliparib) [32,33]. This ability was shown to impact the cell killing effects in HepG2 cells with a single dose of Talazoparib having a greater impact than that of Veliparib [17]. This proof-of-principle study was supported by another study that investigated the impact of three PARPi: Rucaparib (AG014699), Iniparib (BSI 201) and Olaparib (AZD 2281). Several parameters were assessed: cell proliferation (MTT), apoptosis (flow cytometry) and detection of apoptotic protein (by Western blot) and cell migration. All three PARPi significantly inhibited proliferation and migration in the HepG2 cell line. It should be noted that these inhibitors also have a positive effect on apoptosis in this model [34].

Beyond the effects of PARPi used as a single agent, it is interesting to exploit these molecules for their synthetic lethality effect and as attenuators of other therapeutic approaches. As mentioned before, synthetic lethality is defined as the fact that the loss of either of the two genes is not lethal per se but that their concomitant inactivation results in cell death [35]. This concept of synthetic lethality applies in cells that are highly dependent on PARP activity due to, for instance, an HR deficiency, such as in tumor cells carrying BRCA1 mutations, as found in breast cancer. To date, in HCC, no single tumor mutation is predominant enough to warrant the use of PARP inhibitors alone, although it should be noted that based on in vitro models [17], the expression of the HBV HBx protein in HBV-associated HCC may result in a DNA-repair-deficient background that might allow such an approach (discussed below). A number of combined treatments have been evaluated. For example, cotreatment with Chloroquine and the PARPi Niraparib promoted the formation of gammaH2AX foci, a molecular marker of DNA damage, but also inhibited the recruitment of the HR repair protein RAD51 to DSB sites [36]. The PARPi Olaparib could also overcome Sorafenib resistance by remodeling the pluripotent transcriptome in hepatocellular carcinoma [37] and would thus enhance Sorafenib’s ability to eliminate residual HCC tumors and improve the therapeutic efficacy of current Sorafenib therapy.

However, a single-arm trial followed patients with Sorafenib-refractory advanced hepatocellular carcinoma treated with a combination of temozolomide and veliparib and was stopped due to a low objective response rate [10]. This highlights the need to combine PARPi with other therapies. To date, only two clinical trials have been reported on the use of PARP inhibitors in HCC, and, unlike other cancers sites where the number of studies coupled with RT is increasing, no such studies are documented as being in progress in HCC (https://clinicaltrials.gov/, accessed on 29 July 2022) (Table 1). Other combinations of drugs could also be considered. Although the use of immune checkpoint inhibitors (ICIs) in HCC has shown disappointing results [38], the combination of ICIs with PARP inhibitors is also increasingly being developed in other tumors and warrants study for HCC.

As discussed above, RT is not widely used in HCC management; however, there is accumulating evidence that has shown the attenuation of radiation sensitivity by PARPi in HCC models. A pilot study conducted by Guillot et al. demonstrated that PARP inhibition with 10 µM Veliparib over 2 h decreased HepG2 and PLC/PRF/5 cell survival when treated with radiotherapy [21]. Subsequently, it was also shown that a longer exposure of 24 h with 50 nM Talazoparib combined with 2 Gy irradiation on HepG2, PLC/PRF/5 and Hep3B is significantly more cytotoxic than irradiation alone [17].

Given the high proportion of HBV-related HCC (>80% of HCC are related to HBV or HCV infection), it is crucial to understand the impact of PARPi on HBV-infected tumor cells. Decorsière et al. showed that the HBx viral protein associates with the E3 ubiquitin ligase complex DDB1 (DNA-damage binding protein 1) of the host cell to target the Smc5/6 complex for degradation. This degradation is essential to mitigate the transcriptional repression of the HBV genome by Smc5/6 and to stimulate viral replication. It also identified, for the first time, that the Smc5/6 complex is a viral restriction factor [39]. The Smc5/6 complex is implicated in different cellular mechanisms, including the HR DNA repair process and the maintenance of chromosomal integrity, and is also involved in a late step of cellular DNA replication. Human Smc5/6 complex promotes sister chromatid homologous recombination by recruiting the Smc1/3 cohesin complex to DSB [37,39]. These alterations provide a background in tumor cells that would favor the use of PARPi potentially as single agents or in combination with external agents, such as RT, which would generate DNA strand breaks, as outlined above (Figure 1). In order to test this hypothesis, our team treated different cell lines: HepG2 used as an HBV negative control, HepG2.2.15 used as an HBV positive control, as they contain the HBV genome, and finally, HepG2 K6, which contains the HBV genome but lacking the HBx expression. HepG2.2.15 cells treated with 10 µM veliparib for 24 h prior to exposure to 2 Gy of gamma irradiation showed a significant reduction in clonogenic survival compared to HepG2 K6 and HepG2 cells treated under the same experimental conditions. Furthermore, in another HBx-inducible model, Hepa-RG TRX, treatment with 50 nM talazoparib combined with 2 Gy also decreased cell growth compared to irradiation treatment alone [17].

Another independent study confirmed the impairment of DNA double-strand break repair in HBx-expressing HCC cells using a sensitive reporter to monitor HR. Treatment with the PARPi Olaparib was significantly more effective against HBx-expressing HCC cells, and overexpression of Smc5/6 prevented these effects [40]. Altogether, these results suggest that HR deficiency in HBV-associated HCC leads to increased susceptibility to PARPi alone and one that can be attenuated by combining PARPi with radiation treatment [17,39]. Therefore, in vitro potentiation of cell death by PARPi alone or in combination with radiation exposure, taken together with the observations of elevated DNA damage levels in HCC tumor tissues, may represent a vulnerability that can be exploited for therapeutic benefit. Indeed, while RT has not been widely used for HCC treatment, it is a choice that needs further clinical evaluation.

Another strategy would be to combine RT with other treatments simulating DNA breaks, such as Dbaits, which are a kind of DNA damage decoy [41]. Dbaits are short, chemically stabilized, double-stranded DNA fragments of 32 base pairs, marketed by DNA Therapeutics. They mimic a DSB and are recognized by the DNA repair signaling enzymes DNA-PK and PARP1, which are then activated [42,43]. Under these conditions, the signaling of “real” chromosome damage and recruitment of repair enzymes to the site of chromosomal DNA damage cannot occur. Thus, a DNA damaging treatment, delivered after the administration of Dbait molecules, leads to unrepairable DNA breaks because of the hijacking and/or depletion of repair activity in the cells during the treatment period.

One of the remarkable properties of Dbait is its lack of toxicity on non-tumor cells. In fact, the damage signal created by Dbait does not affect ATM and ATR kinases that control cell-cycle arrest and apoptotic death in the presence of damage. As a result, Dbait-treated primary cells stop dividing in the presence of unrepaired damage and resume dividing once the Dbaits have disappeared and the repair is complete. In contrast, tumor cells lose their ability to arrest during the G1/S transition, notably through the mutation of p53 [44,45,46], which is frequently mutated in HCC tumors, particularly in the context of AFB1-induced liver tumors [47].

## 5. Conclusions

In addition to having a crucial role in DNA SSB repair, PARP1 is involved in several signaling pathways, including HIF, NF-κB or β-catenin. As such, it represents a potential lead in elucidating the mechanisms of HCC carcinogenesis. Thus, the use of PARPi that prevent the repair of damage caused by other drugs or therapies, such as hypofractionated sterotactic body RT, would represent an important lever in the treatment of HCC. It is also possible to consider that these combinations could include RT but also radiosensitizing drugs, such as emodin or lupeol. Although there are many studies of PARPi in other cancers, such as HR-deficient ovarian, breast and pancreatic cancers, preclinical and clinical studies of PARP inhibitors in combination with other therapies in HCC are crucial. The combination of PARP with other therapies seems necessary, as no preferential mutations have been detected in HCC so far. However, the high proportion of HCC patients with viral etiology, with the expression of the viral protein HBx, leading to multistep degradation of Smc5/6 proteins involved in DSB repair, may mean that these patients receiving dual therapy (e.g., PARPi + RT) respond much better than HCC patients with other liver cancer etiologies.

In conclusion, PARP is an indispensable element in the advancement of future therapies for HCC, whether by inhibition or by decoys.

## Figures and Tables

**Figure 1 cancers-14-03806-f001:**
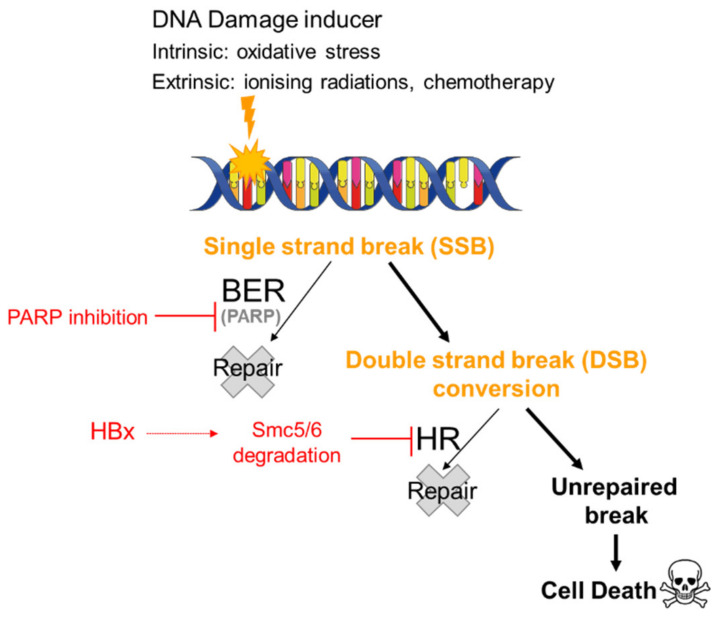
Synthetic lethality of PARP inhibitors in HBV-infected cells. It has been hypothesized that in a patient infected with HBV treated with PARP inhibitors, SSBs generated either intrinsically by oxidative stress or extrinsically by chemotherapy or radiotherapy remain unrepaired and are converted to DSBs. The impact of the degradation of the Smc5/6 complex by HBx on the HR pathway would result in the persistence of these DSBs and cell death.

**Table 1 cancers-14-03806-t001:** Table of completed or ongoing clinical trials of PARP inhibitor combinations with radiotherapy in different cancers compared with clinical trials of PARP inhibitors in HCC. The high number of clinical trials of PARP inhibitors in combination with radiotherapy in other cancers highlights the interest in studying strategies that combine multiple therapies, such as PARP inhibitors and radiotherapy for the treatment of HCC (https://clinicaltrials.gov/ accessed on 29 July 2022).

PARP Inhibitor	Combination Therapy	Condition	Phase	Status	Trial Number
Veliparib	Temozolomide	HCC	II	terminated	NCT01205828
Veliparib	Temozolomide	HCC	I	completed	NCT00526617
Olaparib	Temozolomide, Radiation	Malignant Gliomas	I/II	recruiting	NCT03212742
Niraparib	Dostarlimab, Radiation	Breast Cancer	II	recruiting	NCT04837209
Olaparib	Radiation	Breast Carcinoma	II	recruiting	NCT03598257
Olaparib	Durvalumab, radiation	Pancreatic Cancer	I	Not yet recruiting	NCT05411094
Olaparib	Radiation	Breast Carcinoma	I	completed	NCT02227082
Veliparib	Capecitabine, radiation	Rectal Cancer	I	completed	NCT01589419
Iniparib	Radiation	Brain Metastases	I	terminated	NCT01551680
Niraparib	Radiation	Metastatic Carcinoma of the Cervix	I/II	recruiting	NCT03644342
Olaparib	Radiation	Laryngeal and oropharyngeal carcinoma	I	active, not recruiting	NCT02229656
Veliparib	Radiation	Breast Cancer	I	completed	NCT01477489
Veliparib	Temozolomide, Radiation	Malignant Glioma	II	active, not recruiting	NCT03581292
Olaparib	Radiation, and Immunotherapy	Lung Cancer	I/II	recruiting	NCT04728230
Niraparib	Radiation	Breast Cancer	I	recruiting	NCT03945721
Olaparib	Radiation	Prostate Cancer	I/II	recruiting	NCT03317392
Olaparib	Radiation	Somatostatin receptor positive tumours	I	recruiting	NCT04375267
Olaparib	Pembrolizumab, cisplatin, and radiation	Carcinoma of Head and Neck	II	Not yet recruiting	NCT05366166
Olaparib	Radiation	Breast Cancer	I	Active, not recruiting	NCT03109080

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
