# Peer review of "Poly(ADP-Ribose) Polymerase Inhibition as a Promising Approach for Hepatocellular Carcinoma Therapy"

_cancers, 2022, doi:10.3390/cancers14153806_

Round 1

Reviewer 1 Report

The authors first explained the common causes of HCC and presented the difficulty of treatment in the Late Stage. Based on these, they stated that requires new medicines to improve the prognosis of patients. In treatment, HCC is certainly tolerant of RT as a characteristic, and the need for RT as an option is not high. However, the authors have proposed that the DNA repair mechanism would be working by RT and that be expected synthetic lethality using PARPi in HCC. This can be well understood by the authors' explanation of the molecular biological function of the PARP genes using various citations in this review. From these basic research and clinical viewpoints, the authors stated that the possibility of an inhibitor of PAPR, is beginning to attract attention. The attracting attention is well-understanding because the authors well-explained that the concept of using PARP1 inhibitors is being demonstrated in preclinical breast cancer studies, and explained the FDA and others have approved the use of some cancers.

This review explains in detail the molecular biological findings of PAPR and the current state of clinical application and is expected to deepen the reader's understanding. On the other hand, we are a little worried about "A Novel Approach" in the title. Considering that various drugs of PARPi are already used in various cancers, I think there is a more suitable title.

Author Response

Thank you for your overall positive remarks.

The reviewer underlined the fact that “novel approach” for PARPi use was not totally appropriate.” We are a little worried about "A Novel Approach" in the title. Considering that various drugs of PARPi are already used in various cancers, I think there is a more suitable title

Reply: Thank you for your remark. We removed the term “Novel” and have changed the manuscript title to “ Poly(ADP-ribose) polymerase inhibition as a promising approach for hepatocellular carcinoma therapy”

Reviewer 2 Report

Poly adenosine diphosphate-ribose polymerase (PARP) inhibitors represent an emerging therapeutic strategy in patients harbouring breast-related cancer antigens (BRCA) mutated or homologous recombination-deficient (HRD) malignancies. The family of PARP enzymes consists of 17 nucleoproteins divided into four main groups, the first of which includes PARP1, PARP2, and PARP3 who play a key role during DNA repair. PARP enzymes are involved in detecting single-strand DNA breaks (SSB) and their activation triggers DNA repair factors, such as base excision repair system. In HRD cells (e.g., BRCA1 and BRCA2 mutated), this damage is converted into a double-stranded break (DSB) making DSB unrepairable and leading to selective cell death. Olaparib was the first PARP inhibitor introduced in clinical practice and its efficacy in prolonging outcomes and the manageable safety profile has allowed to test this molecule as a maintenance treatment for several advanced malignancies. Olaparib and other inhibitors are currently under assessment also in HCC, especially as part of combinatorial strategies.

Based on these premises, the study assesses a current, timely topic.
We recommend some changes:
- A linguistic revision would be needed.

- A table summarizing clinical trials in this setting should be included.

- Figure 1 should be removed, we think it is outside the scope of the paper.

- Immune checkpoint inhibitors (ICIs) including pembrolizumab, nivolumab, durvalumab, atezolizumab, etc. have been recently evaluated in HCC patients, and clinical trials assessing single-agent ICI have reported disappointing results. Conversely, immune-based combinations have been more striking. In fact, the phase III IMbrave150 trial assessing the combination of the antiangiogenic agent bevacizumab plus the PD-L1 inhibitor atezolizumab versus single-agent sorafenib has established a new standard of care for HCC patients with advanced disease. According to IMbrave150, atezolizumab - bevacizumab have reported statistically significant and clinically meaningful benefits in several clinical outcomes, including objective response rate (ORR), progression-free survival (PFS), and overall survival (OS), with these advantages also confirmed by the updated results of this trial, showing a median OS of more than 19 months in HCC patients receiving the immune-based combination. Despite ICI seem to have finally found their role in HCC as part of combinatorial strategies, several questions remain unanswered. Among these, the lack of validated biomarkers of response represents an important issue since only a proportion of HCC patients benefit from immunotherapy. Based on these premises, a greater understanding of the role of potential biomarkers including programmed death ligand 1 (PD-L1) expression, tumor mutational burden (TMB), microsatellite instability (MSI) status, gut microbiota and several others is fundamental. In addition, clinical trials on HCC immunotherapy widely differed in terms of drugs, patients, designs, terms of study phases, and inconsistent clinical outcomes. The background of the changing scenario of medical treatment in HCC should be better discussed, and some recent papers regarding this topic should be included (PMID: 34431725 ).

Major changes are necessary.

Author Response

1. "A linguistic revision would be needed”

Reply: Even if one of the co-author is “native English” we undergo a carefull re-reading of the paper.

2. A table summarizing clinical trials in this setting should be included.

Reply: We thank the reviewer for this suggestion and we have added table number 1 entitled “Table of completed or ongoing clinical trials of PARP inhibitor combinations with radiotherapy in different cancers versus clinical trials of PARP inhibitors in HCC”. This clarifies and highlights the value of therapy combination in HCC

3. “ Figure 1 should be removed, we think it is outside the scope of the paper.

We did remove the Figure 1.

4. Immune checkpoint inhibitors (ICIs) including pembrolizumab, nivolumab, durvalumab, atezolizumab, etc. have been recently evaluated in HCC patients, and clinical trials assessing single-agent ICI have reported disappointing results. Conversely, immune-based combinations have been more striking. In fact, the phase III IMbrave150 trial assessing the combination of the antiangiogenic agent bevacizumab plus the PD-L1 inhibitor atezolizumab versus single-agent sorafenib has established a new standard of care for HCC patients with advanced disease. According to IMbrave150, atezolizumab - bevacizumab have reported statistically significant and clinically meaningful benefits in several clinical outcomes, including objective response rate (ORR), progression-free survival (PFS), and overall survival (OS), with these advantages also confirmed by the updated results of this trial, showing a median OS of more than 19 months in HCC patients receiving the immune-based combination. Despite ICI seem to have finally found their role in HCC as part of combinatorial strategies, several questions remain unanswered. Among these, the lack of validated biomarkers of response represents an important issue since only a proportion of HCC patients benefit from immunotherapy. Based on these premises, a greater understanding of the role of potential biomarkers including programmed death ligand 1 (PD-L1) expression, tumor mutational burden (TMB), microsatellite instability (MSI) status, gut microbiota and several others is fundamental. In addition, clinical trials on HCC immunotherapy widely differed in terms of drugs, patients, designs, terms of study phases, and inconsistent clinical outcomes. The background of the changing scenario of medical treatment in HCC should be better discussed, and some recent papers regarding this topic should be included (PMID: 34431725 ).

Reply: Thanks for those remarks.We added more elements on the changing scenarios in HCC treatment and added new references since the landscape in this field is changing fast.

Round 2

Reviewer 2 Report

Acceptance.